# GMDNet: An Irregular Pavement Crack Segmentation Method Based on Multi-Scale Convolutional Attention Aggregation

**Yawei Qi [†] [iD], Fang Wan [†], Guangbo Lei \*, Wei Liu, Li Xu, Zhiwei Ye and Wen Zhou**

School of Computer Science, Hubei University of Technology, Wuhan 430068, China

\* Correspondence: 20000012@hbut.edu.cn

[†] These authors contributed equally to this work.

**Abstract:** Pavement cracks are the primary type of distress that cause road damage, and deep-learning-based pavement crack segmentation is a critical technology for current pavement maintenance and management. To address the issues of segmentation discontinuity and poor performance in the segmentation of irregular cracks faced by current semantic segmentation models, this paper proposes an irregular pavement crack segmentation method based on multi-scale convolutional attention aggregation. In this approach, GhostNet is first introduced as the model backbone network for reducing parameter count, with dynamic convolution enhancing GhostNet's feature extraction capability. Next, a multi-scale convolutional attention aggregation module is proposed to cause the model to focus more on crack features and thus improve the segmentation effect on irregular cracks. Finally, a progressive up-sampling structure is used to enrich the feature information by gradually fusing feature maps of different depths to enhance the continuity of segmentation results. The experimental results on the HGCrack dataset show that GMDNet has a lighter model structure and higher segmentation accuracy than the mainstream semantic segmentation algorithms, achieving 75.16% of MIoU and 84.43% of F1 score, with only 7.67 M parameters. Therefore, the GMDNet proposed in this paper can accurately and efficiently segment irregular cracks on pavements that are more suitable for pavement crack segmentation scenarios in practical applications.

**Keywords:** pavement cracking; semantic segmentation; lightweight model; dynamic convolution; multi-scale convolutional attention



## 1. Introduction

With the ever-accelerating pace of urbanization, pavement construction has become one of the key projects of national and local governments. Pavement cracks are one of the inevitable problems in pavement construction and maintenance, and they seriously affect traffic safety and pavement life. Traditional methods for detecting pavement cracks typically require a significant amount of manual labor and time [1], which are inefficient and prone to missing or misidentifying cracks. Therefore, timely and accurate identification of pavement cracks is crucial for effective pavement maintenance and repair.

Image processing techniques have been widely used in early pavement crack detection. Common image processes include edge detection, morphological operations and filtering. Hu et al. [2] designed a crack extractor on the basis of a simplified locality binary pattern subset, assuming that this subset can extract pavement cracks using information from edges, corners and planar regions, but ignoring the complexities of background textures. Zala-ma et al. [3] employed Gabor filters to extract visible features for crack detection. To overcome the difficulty of parameter selection, they used Ada-boosting to combine several sets of weak classifiers for feature extraction, achieving good results. Li et al. [4] suggested an approach to crack detection using quadratic threshold segmented technology, which used threshold segmentation algorithm to remove pavement markings and perform image segmentation. Shi et al. [5] suggested a pavement crack detection framework called

CrackForest, which applied the structure of random forest to crack detection. The framework introduced a crack description to describe cracks and distinguish them by noise. Although machine vision techniques have been applied for pavement crack detection in various ways, there are also some problems. Because image processing techniques focus only on the local features of images, processing the entire image can result in information loss or incompleteness. For example, when dealing with complex backgrounds and multi-layered cracks, the influence of noise and other factors may lead to inaccurate detection or missed detection. As a result, these methods, which rely on traditional imaging techniques, cannot be used in practice.

In recent years, many investigators began to investigate the application of deep learning techniques to pavement crack detection. Compared to traditional methods, deep learning can better handle the identification of large data and complex features, which enables more accurate detection of pavement cracks [6–8]. Therefore, many researchers have started to apply deep learning methods for the detection of pavement cracks. Cha et al. [9] coupled convolutional neural networks (CNN) with sliding window techniques for crack detection, resulting in significant improvements in validation accuracy and inspection speed. Dorafshan et al. [10] used CNN in combination with edge detection to identify crack images, achieving higher efficiency compared to edge detection methods. Shim et al. [11] used DenseNet [12] as a backbone and streamlined the coding stage by using thin feature maps throughout, resulting in a substantial reduction in parameters. Liu et al. [13] employed the U-Net method for concrete crack identification, which showed higher efficiency and better accuracy compared to Fully Convolutional Networks (FCNs) [14], although some edge smoothing and detail loss issues were observed. Ren et al. [15] suggested a modified deep fully convolutional network called CrackSegNet, which effectively eliminates noise interference and performs end-to-end crack segmentation with complex background crack images. The RDSNet model suggested by Wang et al. [16] features the fusion of detection and segmentation information, which can enhance the detection accuracy to a certain degree. However, the above deep learning-enabled approaches are limited in their ability to identify pavement cracks [17]. Figure 1 shows the segmentation results of these approaches for different pavement cracks, clearly showing the presence of multiple breakpoints in the segmented cracks and poor identification performance when faced with irregular cracks of different scales.

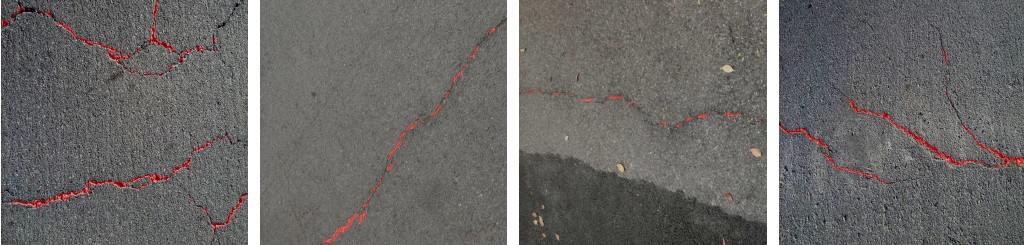

**Figure 1.** Segmentation results of pavement cracks, the red mask portion represents the segmented crack area.

From the research described above, this study suggests a novel network called GMD-Net for irregular crack segmentation in pavements. The main contributions of this study include the following:

1. This study employs and enhances DeepLabv3+ [18]. A lightweight backbone network, GhostNet [19], is utilized to reduce the model's parameter count. By taking into account the diverse scales of crack features, dynamic convolution [20] is employed to improve GhostNet, enabling it to adaptively adjust convolution kernel parameters and enhance feature extraction capability. Additionally, a multi-scale convolutional attention aggregation module (MCAA) is proposed to improve the model's segmentation performance for irregular cracks. A progressive upsampling scheme is introduced

to address the issue of segmentation result discontinuity. Finally, the effectiveness of the proposed module is validated through ablation experiments;

2.  Pavement crack segmentation in practical applications necessitates a lightweight model structure and high-quality segmentation performance. This study conducts a comparative analysis between GMDNet and mainstream semantic segmentation networks using the same dataset to substantiate the superiority of our algorithm in real pavement segmentation scenarios;

3.  This study enriches the Crack500 [21] dataset by incorporating pavement crack images captured by smartphones. The augmented dataset provides a broader spectrum of data, facilitating a more comprehensive evaluation of the model's performance.

The remaining sections are organized according to the following: Section 2 introduces common semantic segmentation models and attention mechanisms. Section 3 presents the structure of GMDNet. Section 4 conducts ablation experiments on GMDNet and provides visual comparisons of segmentation results with other networks. Finally, Section 5 sums up the overall contributions of this paper.

## 2. Related Work

In this section, a comprehensive investigation was conducted on commonly used semantic segmentation models and attention mechanisms, revealing their close relevance to the research objectives of this paper. Understanding the advantages and limitations of these commonly used semantic segmentation methods allows for a better grasp of their characteristics. Moreover, attention mechanisms aid in focusing the model on critical regions, thereby improving segmentation accuracy and robustness. The subsections below describe in detail their working principles and distinctive features. By elucidating these methods, readers will gain a comprehensive knowledge background that forms the foundation for the proposed semantic segmentation models in the subsequent sections.

### 2.1. Semantic Segmentation

In computer vision detection tasks, semantic segmentation enables fine-grained classification at the pixel level. Currently, mainstream semantic segmentation frameworks adopt an encoder–decoder architectural design [22]. The encoder typically comprises several convolutional layers that extract high-level features from the image while gradually reducing its size. In contrast, the decoder uses operations such as upsampling and transpose convolution to recover low-level features to their initial image scale and to predict the category of each pixel. Additionally, links between the encoder and decoder can transmit high-resolution features, which helps to overcome the problem of detail loss that occurs during the decoding process.

The core idea of the Fully Convolutional Network (FCN) [14] is to replace the fully connected layers in image classification networks with transpose convolutional layers and thus recover the feature map size, thereby achieving pixel-wise classification. U-Net [23], proposed for medical cell segmentation. It essentially is made up of a sequence of successive convolutional layers and downsampling layers, which capture contextual semantic information through a contractional path. During decoding, horizontal connections from the encoder are utilized to upsample deep-level and shallow-level features for accurate segmentation mappings. For capturing rich contextual information and improving segmentation performance in complex regions like boundaries, Chen et al. [24] proposed the Deeplab network. It introduces a new convolutional operation using dilated convolutions with upsampling filters, which expands the sensing area to pick up more contextual data with no increase in computational complexity. In addition, this network incorporates conditional random fields to enrich the segmentation data to capture finer details. Based on this, the researchers further proposed Deeplabv2 [25] to derive multi-scale features of the targets. This method utilizes the atrous spatial pyramid pooling (ASPP) module, which detects feature maps from dilated convolutions with distinct sampling rates for multi-scale information. Subsequently, DeepLabv3 [26] developed an encoder–decoder structure using

dilated convolutions for clearer object delineations, and used depth-separable convolutions for improved computational efficiency. Finally, Chen et al. [18] introduced the Deeplabv3+ network model through the addition of a simple and effective decoder module to extend Deeplabv3 and improve segmentation performance. The Deeplab series of networks have achieved satisfactory segmentation results through a series of optimizations, and have become one of the mainstream networks for semantic segmentation.

### 2.2. Attention Mechanism

In visual detection challenges involving object detection, classification and segmentation, the attention mechanism is commonly utilized. In semantic segmentation, the attention mechanism is useful for improving the focus brought by convolutional neural networks on important regions or features. Adding the attention mechanism to neural networks allows more weight to be given to significant features, further enhancing the network's segmentation performance.

Hu et al. [27] first proposed the SE attention, which applies attention to the channel dimension. It enhances useful feature channels and suppresses irrelevant feature channels by adaptively learning the importance weights between channels. CBAM attention [28] calculates attention weights simultaneously in both the channel and spatial dimensions. It adaptively focuses on the significance of various channels and spatial positions within the image to improve feature representation. Effective Channel Attention [29] proposes the idea of not reducing the channel dimension during the process of channel information interaction. Instead, it uses one-dimensional convolution to allocate channel attention weights and reduce information loss. Wang et al. [30] suggested classical Non-Local attention, which captures global information and long-range dependencies through computing the correlations among all positions of the input feature. DANet [31] introduces the Non-Local idea into both the channel domain and the spatial domain. It performs attention calculations in the channel and spatial dimensions separately to capture feature dependencies in both dimensions.

### 3. Materials and Methods

Pavement crack segmentation is an important task in pavement inspection, as accurate segmentation of cracks in pavement images can effectively guide pavement maintenance and management efforts. In practical applications of pavement crack segmentation, segmentation models need to meet the requirements of the lightweight model architecture and accurate segmentation results. However, existing semantic segmentation models have not been optimized specifically for real-world pavement cracks. This paper proposes an irregular pavement crack segmentation network called GMDNet. In the encoder stage, a lightweight GhostNet [19] is employed as the backbone network for crack feature extraction. In terms of irregular cracks, multi-scale convolutional attention aggregation enhances segmentation performance. In the decoder stage, a novel progressive upsampling scheme is utilized to improve the continuity of pavement crack segmentation. Ultimately, GMDNet achieves efficient and accurate pavement crack segmentation. Figure 2 illustrates the general structure of the GMDNet model.

The encoder stage of GMDNet utilizes an improved GhostNet [19] network model as the primary feature extraction network. The input image undergoes feature extraction through Ghost bottlenecks within the GhostNet model, progressively obtaining low-level features containing enriched semantic content and decreasing degree of details, as well as mid-level and high-level features. To further enhance the semantic information in the high-level features, the high-level features are passed through the multi-scale convolution attention aggregation (MCAA) module for multiscale feature aggregation. This yields feature blocks with rich semantic information, which serve as input for the decoder.

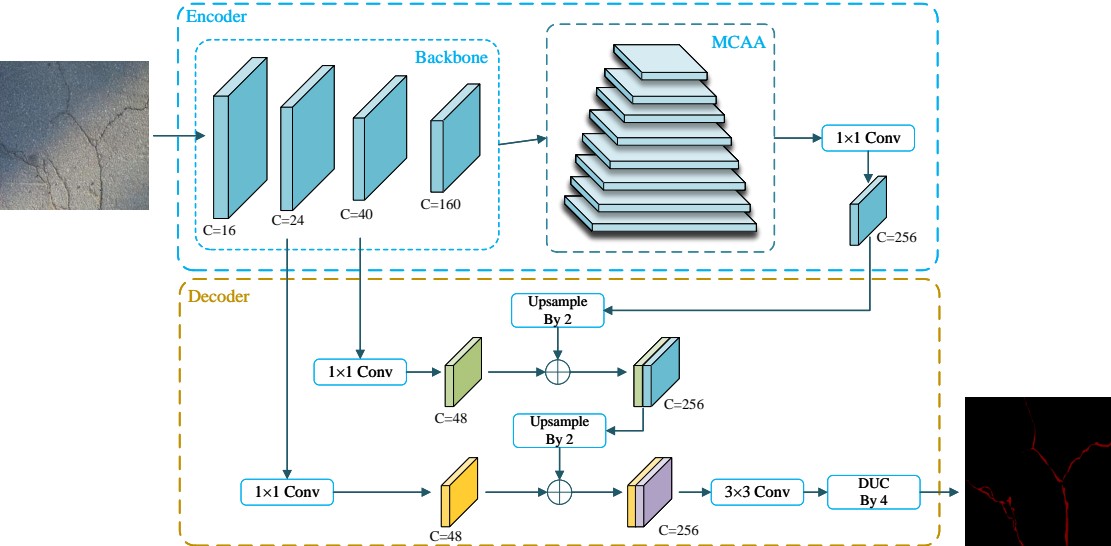

**Figure 2.** The general structure of the proposed GMDNet. Backbone represents the improved Ghost-Net, MCAA represents the multi-scale convolutional attention aggregation module, DUC represents dense upsampling convolution, ⊕ represents the concatenation operation, and C represents the number of channels in the current feature map.

In the decoder stage, GMDNet adopts a progressive upsampling structure. This structure upsamples by progressively integrating low-level detail information into high-level semantic feature maps, while considering both global and detail features. Specifically, the high-level semantic feature maps from the MCAA module are adjusted in channels and then upsampled to the appropriate size. They are then concatenated with the middle-level feature maps for feature fusion, followed by the same operation with the low-level feature maps to merge the features. Lastly, a $3 \times 3$ convolution is used to refine the feature information, and spatial information is restored and segmentation results are outputted via dense upsampling convolution (DUC).

The following sections detail the improvements to GhostNet, multi-scale convolutional attention aggregation (MCAA), and dense upsampling convolution (DUC) of the progressive upsampling structure.

### 3.1. Lightweight Feature Extraction Network Design

Pavement crack segmentation technology often does not work well in practice because the model is too bloated. How to lighten the model is the crucial issue to be addressed in the practical application of pavement crack segmentation algorithms. In convolutional neural networks, the deep feature map has strong semantic information, which is important for accurate crack and background discrimination. However, obtaining deep feature maps requires multiple layers of convolutional operations, a process that can create redundancy of information and thus increase the computing effort of the model. Therefore, many optimization methods [32–34] for deep neural networks were suggested to reduce computational complexity and improve segmentation accuracy. A common approach is to use lightweight feature extraction networks or introduce attention mechanisms to reduce unnecessary computation and information redundancy while maintaining model validity.

GhostNet [19] is a lightweight neural network structure proposed by Huawei Noah's Ark Laboratory in 2020. Compared with other popular lightweight network structures, GhostNet stands out in terms of computational performance. For a significant volume of convolutional computation in deep neural networks, the Ghost module in the GhostNet network starts from the problem of feature map redundancy and exploits the similarity of feature maps to obtain a rich feature map with less computation. Although GhostNet has achieved excellent performance in extracting features from regular objects, the model's

feature extraction capability needs further improvement for the complex and variable pavement cracks in realistic environments. According to the literature [20], dynamic convolution allows adaptive adjustment of convolution kernel parameters based on the input image. Therefore, this paper rethinks the structure of the Ghost module and improves it using dynamic convolution. Figure 3 illustrates the modified Ghost module [19].

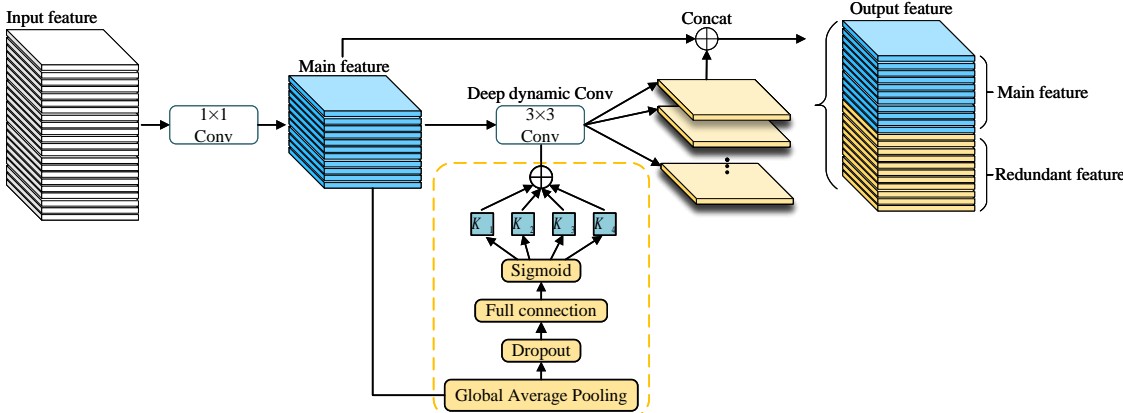

**Figure 3.** The overall structure of the improved Ghost module, which is the main building block of GhostNet.

Specifically, the linear transformation in the Ghost module divides the input feature into two branches. The first branch performs a standard convolution operation, using $1 \times 1$ convolution to achieve information interaction between channels and extract the dominant feature layer from input feature. The second stage processes dominant features layer by layer through linear transformation to produce similar feature maps. By concatenating the above two types of feature layers, the final output feature layer will be generated. The improvement made in this paper introduces dynamic convolution into the process of generating similar feature maps in the second branch. Dynamic convolution does not increase the width and depth of the neural network. It can adjust the convolution kernels based on the features of the input data to better capture the crack features. In dynamic convolution, the convolution kernel parameters are obtained by weighted computation of four identical convolution kernels, and the weights of the four kernels are computed from the input feature. For input-dominant feature, the process of dynamic convolution can be described as:

$$Output\ (x) = \sigma((\alpha_1\ K_1 + \alpha_2\ K_2 + \alpha_3\ K_3 + \alpha_4\ K_4\ ) * x) \tag{1}$$

where $\sigma$ represents the activation function, $\alpha_i$ represents the weighted parameters that depend on the input sample, $K_i$ represents each convolutional kernel, and $*$ represents the convolution operation. The calculation process of $\alpha_i$ can be described as follows:

$$\alpha_i\ (x) = Sigmoid(GAP(x)A) \tag{2}$$

where parameter $A$ represents a matrix that maps the dimension of input features to the number of convolutional kernels. $GAP$ stands for global average pooling, which is used to compress the feature layer and obtain global spatial information. $Sigmoid$ represents the activation function used to generate weighted parameters for the four convolutional kernels.

GhostNet [19] consists of a series of Ghost bottlenecks, which are based on the residual structure of ResNet [35] and leverage the Ghost module to achieve efficient compression and information transfer of feature maps. As depicted in Figure 4, the improved Ghost module is utilized to construct Ghost bottlenecks [19], where (a) represents a Ghost bottleneck with a stride of 1, and (b) represents a Ghost bottleneck with a stride of 2. The Ghost bottleneck with a stride of 1 is employed to extract more feature information while maintaining

the size of the feature map. Conversely, the Ghost bottleneck with a stride of 2 compresses the feature map, reducing computational complexity and memory consumption. To strike a balance between feature resolution and feature range, the improved GhostNet, with dynamic convolution, is adjusted to have four downsampling operations as the backbone network for GMDNet for this study. Specific network layer configurations can be found in Table 1.

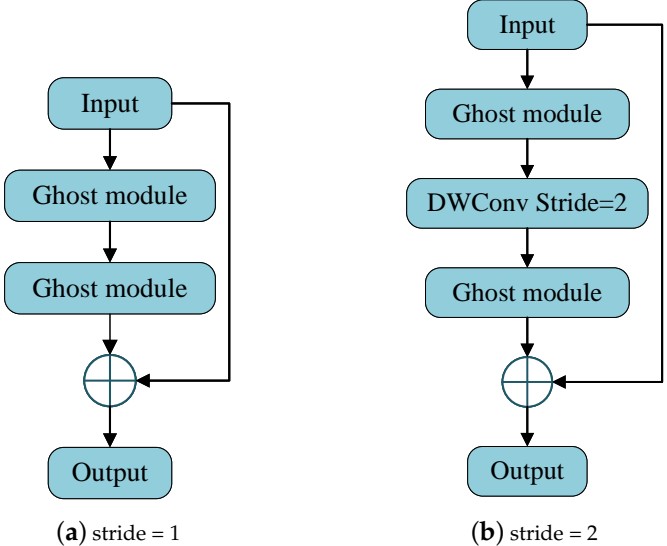

**(a)** stride = 1        **(b)** stride = 2

**Figure 4.** The overall structure of the Ghost bottleneck. $\oplus$ represents the concat operation.

**Table 1.** The specific network structure of GhostNet as a backbone network. Input represents the size and number of channels in the input feature map. G-bneck represents the Ghost bottleneck, exp denotes the expansion size of channels within the Ghost bottleneck, out indicates the number of output channels, and SE denotes whether the SE attention module is utilized.

| Input | Operator | Exp | Out | SE | Stride |
|---|---|---|---|---|---|
| $512^2 \times 16$ | Conv2d $3 \times 3$ | - | 16 | - | 2 |
| $256^2 \times 16$ | G-bneck | 16 | 16 | - | 1 |
| $256^2 \times 16$ | G-bneck | 48 | 24 | - | 2 |
| $128^2 \times 24$ | G-bneck | 72 | 24 | - | 1 |
| $128^2 \times 24$ | G-bneck | 72 | 40 | 1 | 2 |
| $64^2 \times 40$ | G-bneck | 120 | 40 | 1 | 1 |
| $64^2 \times 40$ | G-bneck | 240 | 80 | - | 2 |
| $32^2 \times 80$ | G-bneck | 200 | 80 | - | 1 |
| $32^2 \times 80$ | G-bneck | 184 | 80 | - | 1 |
| $32^2 \times 80$ | G-bneck | 184 | 80 | - | 1 |
| $32^2 \times 80$ | G-bneck | 480 | 112 | 1 | 1 |
| $32^2 \times 112$ | G-bneck | 672 | 112 | 1 | 1 |
| $32^2 \times 112$ | G-bneck | 672 | 160 | 1 | 1 |
| $32^2 \times 160$ | G-bneck | 960 | 160 | - | 1 |
| $32^2 \times 160$ | G-bneck | 960 | 160 | 1 | 1 |
| $32^2 \times 160$ | G-bneck | 960 | 160 | - | 1 |
| $32^2 \times 160$ | G-bneck | 960 | 160 | 1 | 1 |

### 3.2. Multi-Scale Convolutional Attention Aggregation Module (MCAA)

Irregular pavement cracks have always been a significant challenge in pavement crack segmentation, given their diverse shapes and scales. Some cracks are extremely thin, while others are relatively wide. Using a single-scale feature makes it difficult to effectively segment all types of cracks [36]. DeepLabv3+ [18] is a deep learning-based semantic segmentation network that employs an ASPP module with multiple parallel

convolution branches, incorporating dilated convolutions of different scales to capture information at various scales. However, these parallel convolution and pooling operations often merge unrelated pixel regions when dealing with diverse-shaped cracks, resulting in decreased segmentation accuracy. To address these issues, this study draws inspiration from the segnext [37] network and proposes a novel multi-scale convolutional attention aggregation module, as illustrated in Figure 5. Multiple parallel convolution branches are used to aggregate more comprehensive and rich crack features, enabling the model to accurately identify and segment crack regions in the image. The detailed implementation approach of this module will be described below.

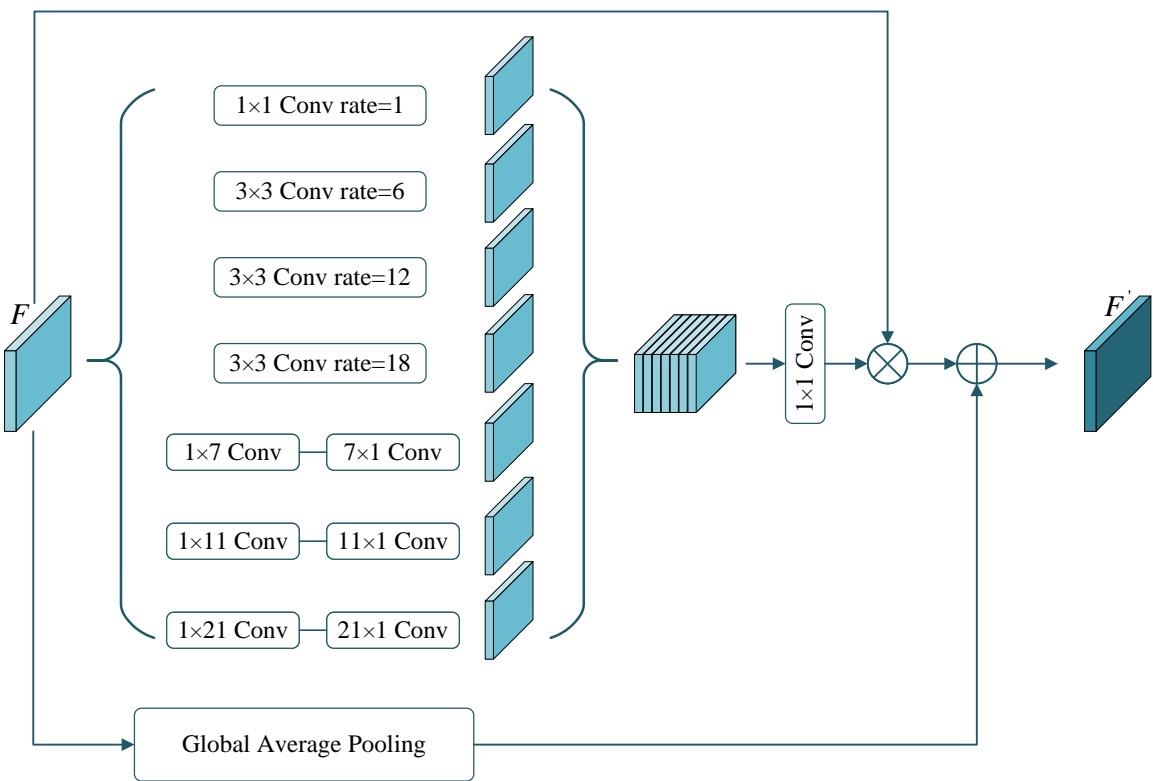

**Figure 5.** The overall structure of MCAA. $\otimes$ represents the attention weight calculation, $\oplus$ represents the concat operation.

The multi-scale convolutional attention aggregation (MCAA) module mainly consists of two branches: the multi-scale feature extraction branch and the global average pooling branch. The multi-scale feature extraction branch employs four square kernel convolutions with dilation rates of 1, 6, 12, and 18, as well as six stripe kernel convolutions, to capture multi-scale feature information. The use of dilated convolutions expands the network's receptive field without increasing additional computational costs. Setting various dilation rates allows acquisition of various reception areas, thereby capturing contextual information on cracks at different scales. In designing the structure of the MCAA module, this paper concatenates two stripe convolutions to mimic standard convolutions [37]. This approach reduces the computational cost and enhances the perception of stripe features, which is beneficial for crack segmentation scenarios.

In order to aggregate the contextual information of cracks of different shapes, the module fuses the six feature maps obtained from the multi-scale feature extraction branch, interacts the information across channels by $1 \times 1$ convolution, and the output is directly used as the attention weight to weight the original features extracted from the $1 \times 1$ convolution branch pixel by pixel, and finally, the weighted feature map is stitched with

the globally averaged pooled feature map as the output feature map $F'$. The multi-scale convolutional attention aggregation module is computed as follows:

$$F' = F \bigotimes Conv_{1 \times 1} \left( \sum_{i=1}^{7} Scale_i(F) \right) \bigoplus GAP(F) \tag{3}$$

where $F$ indicates an input feature, $F'$ indicates an output feature, $\otimes$ is the element-by-element matrix multiplication operation in $Scale_i$, $i \in \{0, 1, 2, 3\}$ denotes the $i$-th branch in the multi-scale feature extraction branch, $\oplus$ is the feature merging operation, and $GAP$ is the global average pooling.

The multi-scale convolutional attention aggregation module achieves feature extraction for different scales by introducing multiple convolutional kernels of different sizes and setting different Dilated Convolution expansion rates. The introduction of banded convolution can compensate for the deficiency of traditional convolution in handling features at different scales, and further improve the model's feature perception at different distance ranges. It combines the multi-scale information of traditional convolution and banded convolution, which can provide better feature extraction and expression when dealing with complex and variable pavement crack shapes, and thus has a better ability to segment irregular pavement cracks.

### 3.3. Dense Upsampling Convolution (DUC)

For pavement crack segmentation, conventional upsampling techniques can lead to indistinct and fragmented crack boundaries in the segmented results. To obtain high-resolution semantic feature maps, convolutional neural networks typically capture deeper semantic information by performing downsampling operations. However, this approach also decreases the quality of the feature maps, resulting in a lack of intricate space data in images. Deep feature maps are spliced with shallow feature maps in certain networks, gradually restoring the resolution to match the high-resolution size of the original map through the utilization of up-sampling techniques like bilinear interpolation. However, simple bilinear interpolation alone cannot solve this problem, because deep semantic features lose their positional alignment with shallow features after padding, convolution, and other operations.

Dense upsampling convolution (DUC) is a structure depicted in Figure 6 [38]. It upscales the downsampled feature maps to the target resolution by learning a series of magnified filters. If the initial image size is $H \times W$ and the downsampling factor during the coding phase is $d$, then the input feature map size for DUC is $\frac{H}{d} \times \frac{W}{d} \times C (h \times w \times C)$. First, a $1 \times 1$ convolution is performed to generate feature maps of size $(h \times w \times d^2) \times L$, which can be understood as $L$ feature maps of size $h \times w \times d^2$. Each feature map of size $h \times w \times d^2$ is transformed into $H \times W \times 1$, which represents a label map for a category. During this process, DUC uses learnable parameters to generate magnified filters. There are $L$ categories in total, and they are combined to form the label map of size $H \times W \times L$ for all categories. The central concept for DUC involves dividing the entire label map into subsets of the same size as the input feature map, i.e., to transform the whole label map into smaller multichannel label mappings. This conversion can be applied directly through convolution operations on both feature maps, with no need to insert additional values as in deconvolution. Therefore, while training the neural network, DUC can adaptively learn how to restore missing detailed information, effectively resolving the problem with segmentation discontinuity for pavement crack segmentation results.

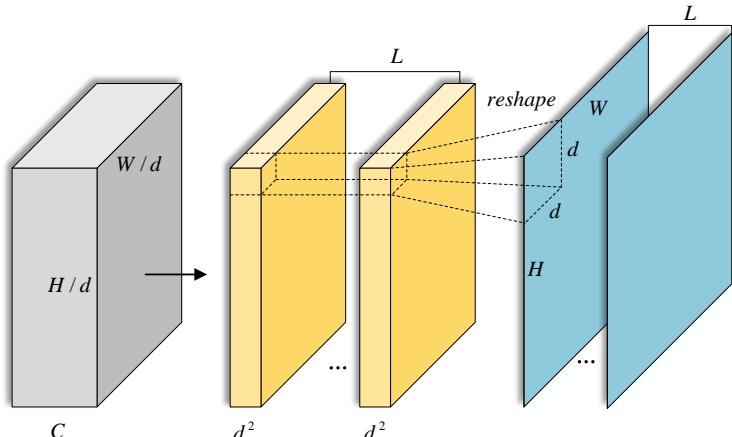

**Figure 6.** The overall structure of DUC.

## 4. Experiments and Results

This section introduces the dataset and experimental setup utilized in this study. Next, a description is presented for the conducted ablation experiments aimed at validating the effectiveness of the proposed GMDNet model. These experiments involve quantitatively analyzing each module to demonstrate their effectiveness. Lastly, a comparison is conducted between the GMDNet model presented in this paper and mainstream semantic segmentation models. The comparison is performed using the Gaps384 and HGCrack datasets, with the aim of demonstrating the efficiency of the proposed pavement crack segmentation model.

### 4.1. Data Collection

The Crack500 [21] dataset is an open-source public dataset that consists of 500 high-resolution images depicting cracks in cement concrete pavement. These images were taken with a digital camera from various distances at different angles, encompassing diverse lighting conditions, angles, and surface textures of the pavement. To overcome the limited image count in the Crack500 dataset and assess the algorithm's generalization ability effectively, this study supplemented the dataset by collecting pavement crack data on campus using smartphones of the same resolution. Consequently, a new dataset named HGCrack was acquired. Figure 7 illustrates a selection of pavement crack images collected on campus.

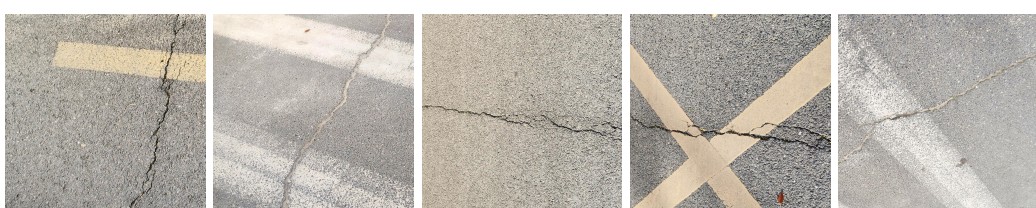

**Figure 7.** Pavement crack image data collected on campus.

The HGCrack dataset includes diverse crack data that has been added to simulate complex pavement crack scenarios in practical applications, based on the Crack500 dataset. Prior to conducting experiments with the HGCrack dataset, various data augmentation methods, including random cropping and flipping, were employed to broaden the dataset. Ultimately, we obtained 2000 images of pavement cracks, along with their corresponding segmentation masks. Among these, 1400 images were designated as the training set, 400 images as the validation set, and 200 images as the test set. Several sample examples from the HGCrack dataset are presented in Figure 8.

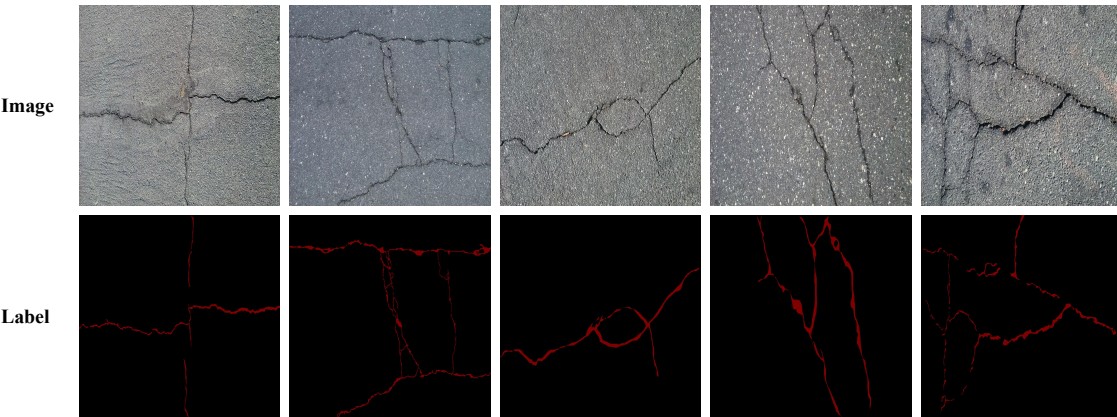

**Figure 8.** Some images and labels in the HGCrack dataset.

To further validate the effectiveness and generalization ability of the GMDNet model, this study chose the publicly available benchmark dataset Gaps384 [39] for testing in the field of pavement crack segmentation. Gaps384 comprises 384 pavement crack images with a resolution of $1920 \times 1080$. These images primarily consist of small and irregular cracks, thereby increasing the difficulty of crack segmentation and facilitating a more comprehensive evaluation of the model's performance. Figure 9 depicts a subset of sample images from the Gaps384 dataset.

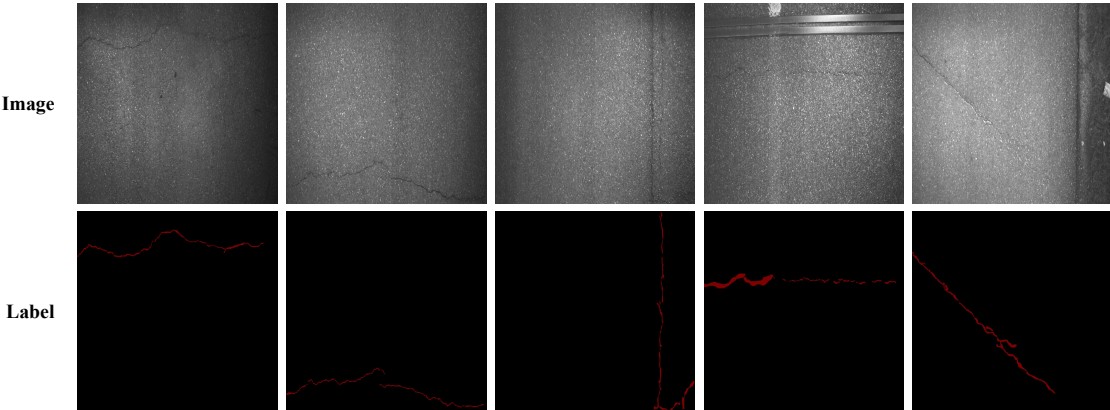

**Figure 9.** Some images and labels in the Gaps384 dataset.

### 4.2. Experimental Setup

#### 4.2.1. Experimental Platform and Training Details

The proposed model GMDNet is implemented using the PyTorch deep learning framework. To ensure consistent dataset quality, the image resolution is uniformly adjusted to $512 \times 512$ as the network input before training. The training GPU is the NVIDIA GeForce RTX3060 (12 GB), the CPU is the Intel Core i7-11700 @ 2.50 GHz, and the memory size is 32 GB. To ensure effective training, this experiment adopts transfer learning by loading the pre-trained weights of the backbone network for accelerating model convergence. The training uses the Adam optimizer with adaptive capability, and the initial learning rate is set to $5 \times 10^{-4}$. Due to GPU memory limitations, the batch size is set to 8, and we train for a total of 100 epochs.

#### 4.2.2. Evaluation Criteria

In this study, the following criteria were used to evaluate the accuracy and effectiveness of the proposed GMDNet for pavement crack detection: MPA, MIoU and F1 score. Higher values of these metrics indicate superior overall network performance [40]. The complexity

of the model is evaluated based on the values of Params and FLOPs. Lower values of these metrics indicate a more lightweight network. Params represents the number of parameters within the model, where a higher number usually indicates a more complex model. FLOPs quantifies the floating-point operations executed by the model during inference. Higher FLOPs indicate greater computational resources and time required for inference. Other performance evaluation metrics are calculated through the confusion matrix presented in Table 2. Detailed calculation procedures for MPA, MIoU and F1 Score score are presented below. The detailed calculation procedures for MPA, MIoU, and F1 score are outlined below.

**Table 2.** All the results of pavement crack prediction.

| Predicted / Ground Truth | Pavement Crack | Background |
|---|---|---|
| Pavement Crack | True positive (*TP*) | False positive (*FP*) |
| Background | False negative (*FN*) | True negative (*TN*) |

*MPA* stands for mean pixel accuracy, indicating percentage correctly categorized pixels for every category, which is subsequently accumulated and averaged. The formula is as follows:

$$MPA = \frac{1}{K+1} \sum_{i=0}^{k} \frac{TP + TN}{TP + FP + TN + FN} \tag{4}$$

*MIoU* can be understood as the average intersection over union (IoU) between predicted and ground truth regions of various pixels, which reflects the overlap between the segmented and actual targets. The formula is as follows:

$$MIoU = \frac{1}{K+1} \sum_{i=0}^{K} \frac{TP}{FN + FP + TP} \tag{5}$$

*F*1 *Score* represents a weighted mean of precision and recall, accounting for both comprehensiveness of precision as well as accuracy of recall. The formula is as follows:

$$precision = \frac{TP}{TP + FP} \tag{6}$$

$$recall = \frac{TP}{TP + FP} \tag{7}$$

$$F1\ Score = 2 \times \frac{precision \times recall}{precision + recall} \tag{8}$$

In the above formula, *TP* represents the amount of pixels that are truly labeled as cracks and correctly segmented as cracks. *FP* is the number of pixels truly labeled as background that are incorrectly segmented as cracks. *TN* is the number of pixels truly labeled as background and correctly segmented as background. *FN* is the number of pixels that are truly labeled as cracks but incorrectly segmented as background [41]. Here, $k + 1$ represents the number of categories (there are two categories in this study, cracks and background, where $k$ is set to 1).

### 4.2.3. Training Process Evaluation

The binary cross-entropy is adopted as the loss function for training, since the semantic segmentation of pavement cracks involves a classification problem with only two categories-cracks and background. The binary cross-entropy loss function formula is:

$$L_{bce} = -\frac{1}{N} \sum_{i=0}^{n} [y_i ln p_i + (1 - y_i) ln(1 - p_i)] \tag{9}$$

where $L_{bce}$ represents the binary cross-entropy loss, $N$ represents all pixels of the image, $y_i$ represents the label values for the $i$-th pixel, and $p_i$ represents the prediction probability value for the $i$-th pixel.

Figure 10 shows loss curves obtained from experiments using the GMDNet model on the HGCrack dataset. Horizontal axis shows the epochs during model training. In each epoch, the model updates and optimizes its parameters based on the training data. The vertical axis represents the loss function values. A smaller loss function value indicates a better fit of the model's predicted results to the actual labels on the training data. The curves in the graph correspond to the train loss and val loss, respectively. The train loss is computed using the training data during model training and is primarily used to guide model optimization and parameter updates for a better fit to the training data. On the other hand, the val loss is calculated using the validation data after the model training process to evaluate the model's generalization ability. In this experiment, both the train loss and val loss decrease as the training epochs increase. In the first training epoch, the train loss is 0.194, and the val loss is 0.154. By the 100th training epoch, the train loss has decreased to 0.019, and the val loss has decreased to 0.032. The loss function experiences a rapid decrease in the early training epochs, gradually stabilizing thereafter. The overall curve trend is favorable, with both the train loss and val loss dropping below 0.05 by the end of training. This demonstrates that the proposed GMDNet network effectively converges on the HGCrack dataset.

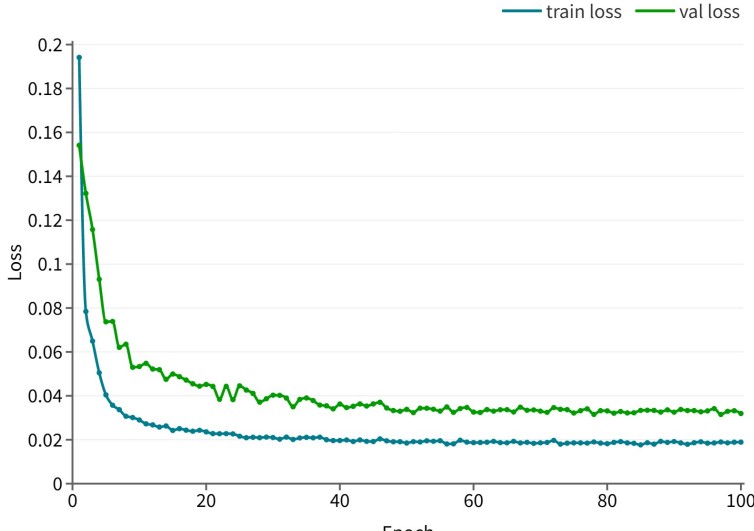

**Figure 10.** The train loss curve and val loss curve obtained from GMDNet experiments on the HGCrack dataset.

### 4.3. Ablation Experiments

To assess the efficacy of the proposed backbone network (GhostNet [19]), multi-scale convolution attention aggregation module (MCAA), and progressive upsampling structure (PU) in the GMDNet model, this study conducted ablation experiments based on the classic DeepLabv3+ [18].The aim was to evaluate the accuracy and computational cost of each module. Using the HGCrack pavement crack dataset, we performed four sets of experiments in the same controlled environment to determine the optimal model. These sets include: Group 1, which involved training solely on the classical DeepLabv3+ network; Group 2, where the backbone network was replaced with the improved GhostNet built upon the classical DeepLabv3+ network; Group 3, in which the ASPP module was substituted with the proposed multi-scale convolution attention aggregation module (MCAA) based on Group 2; and Group 4, where the proposed progressive upsampling structure (PU) was utilized on the foundation of Group 3. The results of these four ablation experiments are presented in Table 3.

**Table 3.** Experimental results for different combinations of modules in GMDNet.

| Group | GhostNet | MCAA | PU | MIoU (%) | F1 Score (%) | Params (M) |
|-------|----------|------|----|----------|--------------|------------|
| 1 | | | | 72.28 | 81.22 | 54.71 |
| 2 | ✓ | | | 70.54 | 80.58 | 5.33 |
| 3 | ✓ | ✓ | | 74.89 | 82.97 | 6.14 |
| 4 | ✓ | ✓ | ✓ | 75.16 | 84.43 | 7.67 |

From Table 3, it can be seen, that compared to the various evaluation indicators of the original network in Experimental Group 1, Experimental Group 2, which replaces the backbone network, has only a slight decrease of 1.74% in the model's MIoU under a significant reduction in model parameters. At the same time, the F1 score still remains at a good level. This indicates that the improved backbone feature extraction network (GhostNet [19]) not only can obtains crack features in an inexpensive computational way but also extracts crack features more accurately with the help of dynamic convolution. Furthermore, it can be observed from the results of Experimental Group 3 that, after introducing the multi-scale convolutional attention aggregation module (MCAA), the model's MIoU has increased by 4.35%, showing a significant improvement. Meanwhile, the model parameters have only slightly increased, indicating that the inclusion of the stripe scale and convolutional attention in this module enables the model focusing on more crack features, resulting in a significant improvement in segmentation accuracy with minimal impact on network computational costs. Experimental Group 4 simultaneously made three improvements. Based on Experimental Group 3, it added a new progressive upsampling structure (PU), which reduces the loss of detailed information compared to the original upsampling method, and achieves better pixel segmentation effects in crack boundary regions. As a result, the model's MIoU reached the highest value of 75.16%, and other performance evaluation indicators were also optimal. In summary, the network with three improvements demonstrates the best overall performance, achieving a balance between model accuracy and computational cost, thus being able to adapt to a wider range of application scenarios while meeting the required accuracy. This experiment fully demonstrates the ability of the proposed improvements in enhancing the segmentation effectiveness of pavement cracks.

*4.4. Performance Comparison*

To objectively and quantitatively analyze this study's proposed crack segmentation model, five mainstream methods for semantic segmentation, specifically FCN [14], U-Net [23], DeepLabv3+ [18], PSPNet [42] and SegNet [43], were chosen for comparison. FCN integrates features from multiple levels through upsampling and skip connections to produce the final output. U-Net uses multiple skip connections to combine multi-scale info, enhancing the accuracy for segmentation. The DeepLabv3+ algorithm employs atrous spatial pyramid pooling modules for analyzing targets across various scale ranges, and subsequently combines shallow features to recover boundary and detail information. PSPNet aggregates context information from various regions to explore the global context. SegNet's decoder achieves nonlinear upsampling by utilizing max pooling indices from the encoder, thus avoiding additional learning costs during the upsampling process. Subsequently, the proposed model in this study is evaluated and compared against the aforementioned five mainstream semantic segmentation models using the HGCrack dataset and the Gaps384 dataset. The effectiveness of the proposed model is validated using the HGCrack dataset, while the generalization ability is assessed using the publicly available benchmark dataset Gaps384.

Table 4 displays the evaluation results from the HGCrack dataset. The GMDNet algorithm attains MIoU, MPA, and F1 scores of 75.16%, 87.29% and 84.43%, respectively. Our algorithm outperforms other algorithms in all three accuracy metrics. Our algorithm demonstrates improvements of 2.24%, 3.06% and 2.52% over the best-performing Seg-Net network and other algorithms [44], respectively. Furthermore, our model exhibits

substantially fewer parameters and lower computational costs compared to the aforementioned algorithms.

**Table 4.** The performance of the proposed model is compared with other methods on the HGCrack dataset.

| Model | MIoU (%) | MPA (%) | F1 Score (%) | Params (M) | FLOPs (G) |
|---|---|---|---|---|---|
| FCN [14] | 70.64 | 79.69 | 79.85 | 49.5 | 57.91 |
| U-Net [23] | 71.36 | 80.61 | 80.29 | 29.06 | 197.76 |
| DeepLabv3+ [18] | 72.28 | 83.60 | 81.22 | 54.71 | 83.42 |
| PSPNet [42] | 69.83 | 79.02 | 79.04 | 48.97 | 178.45 |
| SegNet [43] | 72.92 | 84.23 | 81.91 | 29.44 | 105.99 |
| GMDNet (proposed model) | 75.16 | 87.29 | 84.43 | 7.67 | 19.48 |

Table 5 displays the evaluation results using the Gaps384 dataset. The proposed GMDNet algorithm achieves MIoU, MPA and *F*1 values of 70.96%, 80.23% and 79.75%, respectively. All three accuracy metrics are superior to other algorithms, indicating that our model has better generalization ability in crack segmentation.

**Table 5.** The performance of the proposed model is compared with other methods on the Gaps384 dataset.

| Model | MIoU (%) | MPA (%) | F1 Score (%) | Params (M) | FLOPs (G) |
|---|---|---|---|---|---|
| FCN [14] | 67.62 | 77.07 | 75.52 | 49.5 | 57.91 |
| U-Net [23] | 69.05 | 78.09 | 77.80 | 29.06 | 197.76 |
| DeepLabV3+ [18] | 68.51 | 75.86 | 77.31 | 54.71 | 83.42 |
| PSPNet [42] | 68.64 | 79.78 | 77.53 | 48.97 | 178.45 |
| SegNet [43] | 68.78 | 78.55 | 81.91 | 29.44 | 105.99 |
| GMDNet (proposed model) | 70.96 | 80.23 | 79.75 | 7.67 | 19.48 |

For more immediate comparison and analysis about the segmentation results of each model, we visualize and analyze the segmentation results of the proposed model in this study, along with five other mainstream segmentation models, on the Gaps384 and HGCrack datasets.

Figure 11 shows the visualization of segmentation results on the HGCrack dataset. It can be observed the segmentation performed by the FCN [14] network shows the poorest results, with obvious segmentation discontinuity and incompleteness, and tends to miss segmenting small cracks. U-Net [23] can segment the rough outlines of the cracks, but severe false positives and false negatives occur with the detailed parts to the cracks. DeepLabv3+ [18] and PSPNet [42] networks have difficulties in guaranteeing the continuity of crack edges when segmenting small cracks, leading to partial missed detections. Compared to the previous four models, SegNet [43] shows some improvement in segmentation continuity and completeness, but it still suffers from missed segmentations for some small cracks. The proposed GMDNet model of this study performed best with respect to segmentation completeness as well as segmentation continuity.

Figure 12 shows the visualization of segmentation results on the Gaps384 dataset. It can be observed that mainstream segmentation models exhibit a wide range of missed segmentations. Among them, FCN performs the worst in terms of segmentation. U-Net demonstrates superior segmentation results regarding crack details. The suggested model described in this study shows the fewest missed segmentations and false detections and overall surpasses the other methods for segmentation.

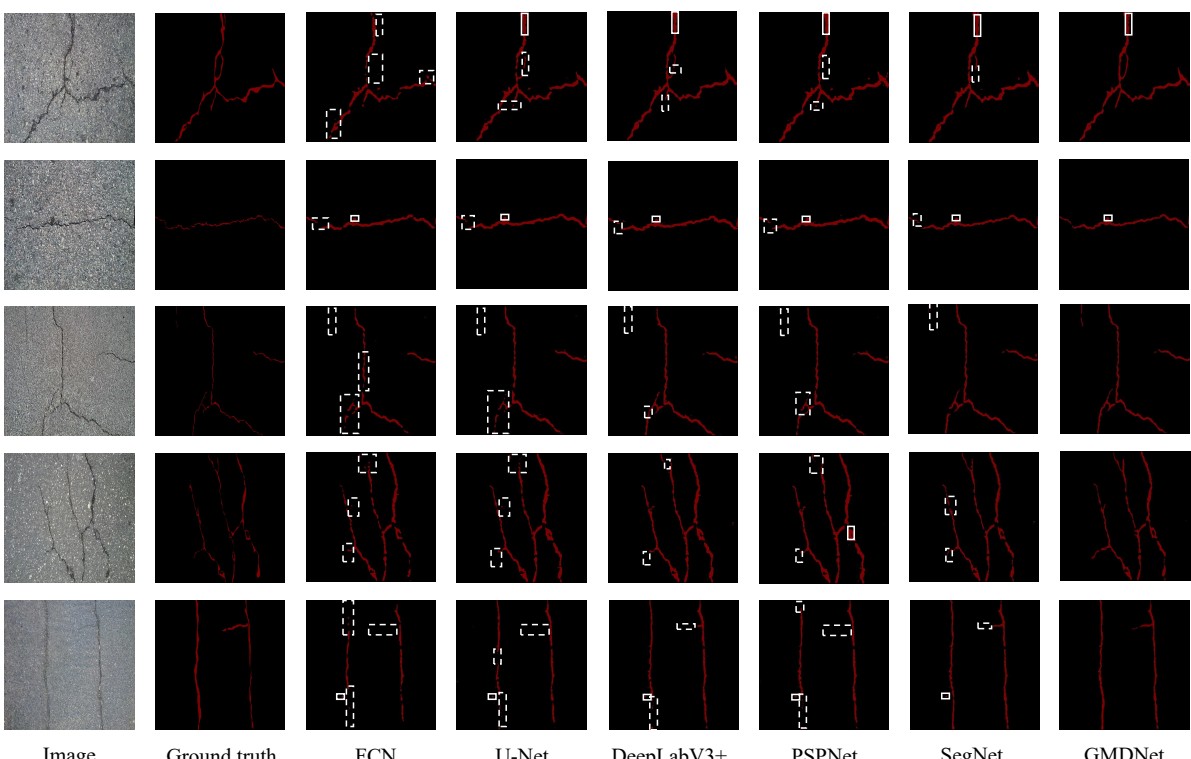

Image    Ground truth    FCN    U-Net    DeepLabV3+    PSPNet    SegNet    GMDNet

**Figure 11.** Visualization results of the segmentation on the HGCrack dataset. Dashed boxes indicate missed detections, while solid boxes represent false detections.

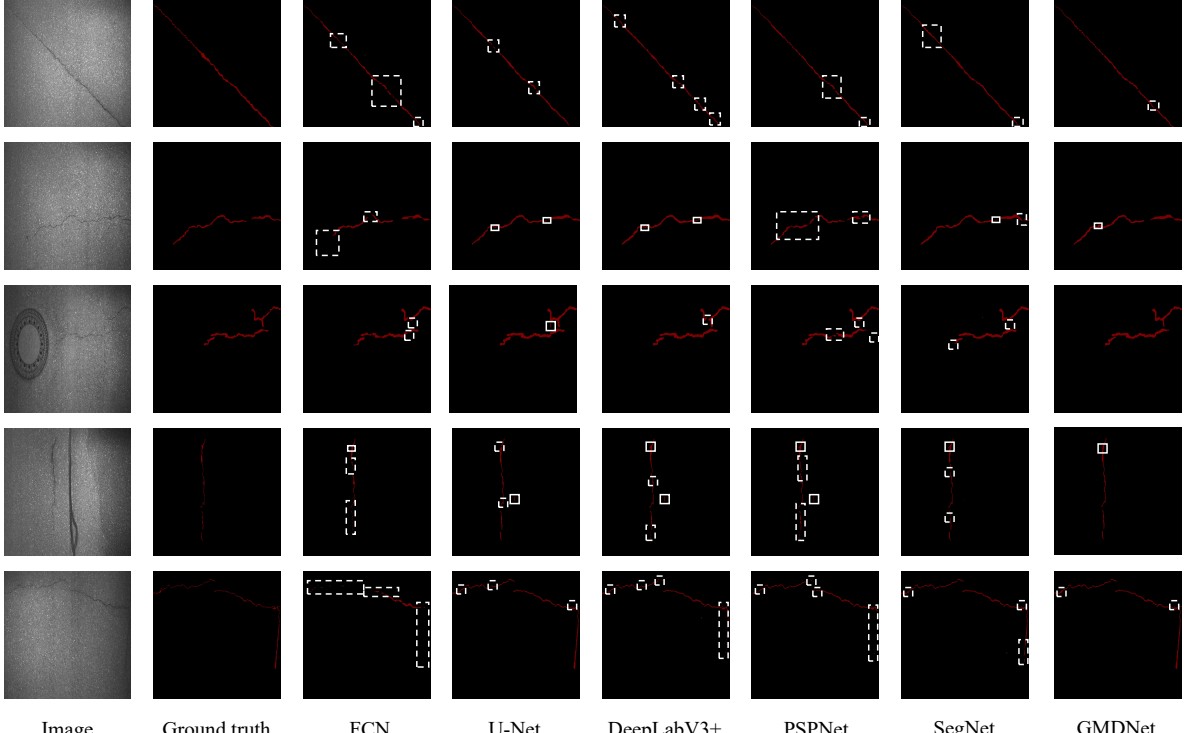

Image    Ground truth    FCN    U-Net    DeepLabV3+    PSPNet    SegNet    GMDNet

**Figure 12.** Visualization results of the segmentation on the Gaps384 dataset. Dashed boxes indicate missed detections, while solid boxes represent false detections.

Overall, the study suggests that the GMDNet model exhibits superior detection rate, contour integrity, and continuity. It generates more refined crack segmentation results

and effectively handles segmentation of irregular cracks. Furthermore, it can accurately and fully segment cracks. Visual comparison tests performed with two datasets have consistently shown that the suggested method is superior to other techniques in pavement crack segmentation.

## 5. Conclusions

This article presents a lightweight multi-scale convolutional attention aggregation segmentation network, which is built upon the enhancement of DeepLabv3+ [18]. In the encoding stage of the network, GhostNet [19] enhanced with dynamic convolution is used as the backbone network, reducing network complexity. By introducing the module for multi-scale convolution attention aggregation (MCAA), the model's focus on crack regions at different scales is enhanced, enhancing the segmentation of irregular cracks. In the decoding stage of the network, a progressive upsampling method is proposed to mitigate the loss of detailed information in the segmented results and address disconnection in crack segmentation. The effectiveness of each suggested module is verified through extensive experimentation and analysis, all of which enhance the model's performance. Finally, the experiment findings using the HGCrack dataset indicate the suggested model offers advancements in pavement crack segmentation, demonstrating significantly improved segmentation results. It achieves an MIoU of 75.16% and an $F1$ score of 84.43% with a parameter size of just 7.67 M. Therefore, the model proposed in this article exhibits excellent effectiveness in pavement crack segmentation with fewer parameters, making it easier to meet practical application requirements. Moreover, the model in this article focuses on improving the segmentation accuracy of small irregular cracks, offering preventive pavement maintenance strategies. For example, pavement maintenance personnel can identify and locate potential problem areas at an earlier stage, preventing cracks from expanding and developing into more severe pavement damage. This is crucial for achieving sustainable pavement maintenance practices and mitigating the risk of traffic accidents.

Although this study has achieved good results in the segmentation of irregular cracks on the pavement surface, further improvement and optimization are still necessary. Firstly, the dataset currently has a limited number of crack scenarios; thus, it is necessary to expand the dataset and gather additional images of pavement cracks with different shapes, sizes, and backgrounds. By including more samples of pavement cracks with complex backgrounds, the model's robustness can be improved, enhancing its accuracy and generalization in a wider range of scenarios. Secondly, the segmentation types of pavement cracks in this study can be further refined, such as linear cracks, grid cracks, block cracks, etc. Therefore, it is necessary to enhance the effectiveness of the model's segmentation for these various crack types and accurately identify and differentiate them in practical applications. Through these efforts, the practical needs of pavement maintenance can be better met, offering a more accurate, stable, and practical solution for pavement crack segmentation.

**Author Contributions:** Conceptualization, Y.Q. and F.W.; Methodology, Y.Q.; Software, Y.Q. and G.L.; Validation, Y.Q.; Formal analysis, Y.Q. and L.X.; Investigation, F.W., G.L. and Z.Y.; Resources, L.X. and W.Z.; Data curation, Y.Q. and G.L.; Writing—original draft, Y.Q.; Writing—review and editing, F.W. and G.L.; Visualization, Y.Q.; Supervision, W.L. and Z.Y.; Project administration, W.Z.; Funding acquisition, L.X. and W.Z. All authors have read and agreed to the published version of the manuscript.

**Funding:** This research was funded by the National Natural Science Foundation of China (Grant No. 62202147) and the Science and Technology Research Project of Education Department of Hubei Province (Grant No. B2021070).

**Data Availability Statement:** Not applicable.

**Conflicts of Interest:** The authors declare no conflict of interest.

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
