# Peer review of "GMDNet: An Irregular Pavement Crack Segmentation Method Based on Multi-Scale Convolutional Attention Aggregation"

_electronics, doi:10.3390/electronics12153348_

Round 1

Reviewer 1 Report

The paper introduces a novel approach, GMDNet, for pavement irregular crack segmentation based on multi-scale convolutional attention aggregation. The authors emphasize the importance of this technology in road maintenance and management. The proposed algorithm leverages several techniques to address segmentation discontinuity and poor performance in irregular crack detection. These techniques include the integration of a ghost module, dynamic convolutional adaptive feature maps, multi-scale convolutional attention aggregation, and a progressive upsampling structure.

The results presented in the paper showcase promising performance on the HGCrack dataset, with a mean Intersection over Union (mIoU) of 75.16% and an F1-Score of 84.43%, achieved using only 7.67M parameters. This indicates that the proposed GMDNet outperforms mainstream semantic segmentation algorithms in terms of both model efficiency and segmentation accuracy.

However, a significant concern with this paper is the lack of detailed information regarding the architecture of the convolutional neural network. Without the necessary specifications, it becomes impossible to reproduce the proposed model and verify the reported results. Replicability is a critical aspect of scientific research, and the absence of architectural details hinders other researchers from building upon this work or applying it to practical scenarios.

To improve the paper before resubmitting, I strongly recommend that the authors provide comprehensive documentation on the architecture of the GMDNet model. This should include detailed descriptions of the network layers, parameter settings, and any other crucial architectural aspects. By doing so, the authors will enable other researchers to reproduce their work, compare their results, and further develop the proposed method.

Additionally, including experiments that demonstrate the effectiveness of the GMDNet model on other datasets, especially publicly available benchmark datasets in the field of pavement crack segmentation, would strengthen the paper's claims. Comparative evaluations against existing state-of-the-art methods would provide a more comprehensive understanding of the proposed algorithm's strengths and limitations.

In summary, the paper introduces an intriguing approach, GMDNet, for pavement irregular crack segmentation. However, the lack of detailed information regarding the architecture of the proposed CNN hinders the reproducibility and validation of the reported results. I strongly encourage the authors to address this issue and provide the necessary architectural details to enhance the paper's scientific value. Furthermore, expanding the experimental evaluation to include additional datasets and comparative analysis would further strengthen the research presented.

The English language is not a big problem.

Reviewer 2 Report

The manuscript presents a multi-scale convolutional attention aggregation-based pavement irregular crack segmentation algorithm (GMDNet) with a lighter model structure and higher segmentation accuracy (when compared to other approaches). The applicability and accuracy of the model was tested and validated using data from the HGCrack dataset. The approach can be of interest to professionals and researchers involved in road pavement maintenance management.

Dear authors, please consider the following comments and suggestions:

Abstract and Keywords

- Page 1, line 1: Consider using “distress” instead of “disease”.

- Page 1, lines 6-11: The paragraph describing the methodology is confusing. Consider revising to make it clearer.

- The keywords "lightweight" and "attention" do not clearly translate what is intended to be conveyed. Expand to clarify meaning.

1.Introduction

- Page 2, figure 1 (and remaining manuscript figures): the figures’ explanation must be presented in the text that precedes the figure. In most cases the content of the figure is explained in the caption or in the text that follows the figure. I advise you to review the location of the text that presents and explains the figures and the content of the captions of these figures. Some captions even incorporate notes, which should not be considered in this way.

- Page 2, figure 1 caption, lines 66, 68, 74, and other manuscript locations: avoid using the form "we" or "our". Review.

- Page 2-3, lines 66-83: The paper's contributions are presented in a format that most closely resembles the conclusions. They should be rewritten and presented in a more synthetic way, highlighting the knowledge gap they fill. I also recommend including a paragraph, at the end of the introduction section, with a summary description of the manuscript organization and content.

2. Related work

- It is not clear why sub-sections 2.1 and 2.2 are presented and how they are related. I recommend preceding the subsections with a framework of their relevance to the purpose of the study and connection.

3. Materials and Method

- Page 5, subsection 3.1: It is not clear to what extent the proposed approach is innovative when compared to the existing ones since the authors refer that other studies have already used the approaches described (eventually in crack segmentation). Clarify this aspect.

- Are figures 3 and following by the authors? If not, include citations.

- Include citations to support the expressions presented throughout the manuscript.

- Page 7, lines 240-243. How are these improvements achieved?  What exactly causes these improvements. Clarify in the text.

In general, it should be clearer what the authors bring to this field of applications/approaches that allow to improve the performance of the proposed model when compared to the existing ones.

4. Experiments and results

- Page 10, line 318: Is the images’ quality captured by smartphones similar to the quality of images of the Crack500 dataset? If not, can mixing images of different quality change uniformity of model performance? Or does the model also seek to overcome this aspect (handling datasets with images of varying quality)?

- Page 12, figure 9: Include an explanation of what the vertical and horizontal axes represent and the difference between the two curves.

- Page 12, table 2 and page 13, table 3: Include the meaning of Params(M) and FLOPs(G).

- Page 13, lines 424-425: Justify why the authors think that an improvement between 2.2 to 3.1% in MIOU, MPA and F1 is significant. Eventually justify with other studies (citations).

- Standardize the acronym: MIoU, MIOU or mIoU

5. Conclusion

Include the main limitations of the approach and experiment, the main experiment results that validate the relevance of the proposed approach and the practical use of the approach, namely in the segmentation/identification of fine cracks (to support a preventive approach to road pavement maintenance). The conclusions should justify why this approach is better that other approaches and more suitable to support pavement maintenance.

I recommend an English review to correct lapses and improve the construction of some sentences.

Round 2

Reviewer 1 Report

The authors have addresses the comments. The article can be accepted in its current form.

The authors have addresses the comments. The article can be accepted in its current form.

Author Response

It is a great honour to have your approval of this work and thank you for improving the quality of the article.

Reviewer 2 Report

The authors have addresses most of the comments and suggestions.

Regarding point and response 6:

The description of the paper's contribution continues to sound more like the conclusions:  the use of expressions such as "… experimental results demonstrate …", "… same dataset reveals that …" or "The findings show that the model ..." are not adequate for this purpose.

Please review.

Note that response to point 8 is more in line with the format intended for a description of the study contribution than the one presented in response 6.

Regarding point and response 7:

It is preferable to use "sections" rather than "chapters".

Finally, I recommend one last review of English. 

I recommend one last review of English. 
